# The Evolutionary System of the Biosphere and the Metameric Concept of Its Evolution: From the Past to the Future

**Alexander Protasov [1] and Sophia Barinova [2],***

[1] Institute of Hydrobiology, National Academy of Science of Ukraine, 12, V. Ivasiuk Avenue, 04210 Kiev, Ukraine; pr1717@ukr.net

[2] Institute of Evolution, University of Haifa, Mount Carmel, 199 Abba Khoushi Avenue, Haifa 3498838, Israel

\* Correspondence: sophia@evo.haifa.ac.il; Tel.: +972-4-8249799

**Abstract:** We offer a detailed description of our previously published new concept of the evolution of the biosphere as an integral system of its states over time, united by development trends. The structure of the biosphere is considered as a hierarchical fractal system, and the elementary unit of the biosphere is an ecosystem. The fractal structure of the biosphere corresponds to the emergent principle: each lower level is an element of a more complex system and has its own properties. The proposed concept of biosphere evolution is based on the general categories of dialectics: states and interstates, continuity and discreteness, reproducibility and uniqueness. The evolutionary history of the biosphere is a metameric picture of changing states and interstates. The most important feature of the biosphere organization in space–time is a complex system of continua. The development of an integral biospheric system occurs in a time continuum: in the biosphere, the differences between the early and subsequent states are quite significant and obvious. Moreover, these differences are associated with fundamental complications, development, which is, in fact, evolution. The states of the biosphere in certain periods are linked by trends that form an evolutionary system. Continuing states, when the system remains self-identical for a long period, are replaced by new states through interstates. Based on the principle of actualism, the problems of the biosphere's future and evolutionary trends of the biosphere under anthropogenic impact are considered.

**Keywords:** evolution; biosphere; fractal structure; development trends; states; interstates; metameric model; continuity





## 1. Introduction

Many global processes (in particular, climate change) are considered primarily from the perspective of changes in human living conditions [1,2] even if the impacts of certain factors on ecosystems are considered [3]. This does not take into account the fundamental position expressed by V.I. Vernadsky: a human, like any other species of living beings, is a product of the biosphere, and they do not exist and cannot exist outside it [4]. Forecasting the future of humans and the biosphere is impossible without knowing the history and regularities of the biosphere's development. The fundamental principles of the biosphere system's structure and functioning, both before and with human participation, have been and remain unchanged.

Vladimir Vernadsky's concept is that the complex of living organisms (he calls it "living matter", Glossary) interacts with the inert elements of the Earth's geological system, actively influences them, and forms the biosphere as a united self-regulating and self-sustaining system. V. Vernadsky's concept of the connections and development of living matter in the changing conditions of the planet as a whole was like a Copernican revolution in understanding the world [5,6]. It allowed for the first time to look at life and the geological basis of life existence as a single whole, a single system, although earlier, such great minds as J.B. Lamarck, A. Humboldt, and E Suess tried to move from particular knowledge to planetary generalizations.

Recently, the idea of the unity of evolutionary processes in nature has been actively discussed [7,8]. Self-development and irreversibility of the general progressive course of evolution are important. The evolution of the biosphere has repeatedly been the subject of scientific consideration and analyses, in general philosophical terms [9–11]. More specific issues related to both the evolution of biotic components of the biosphere and the evolution of bioinert systems have also been discussed [12–16]. Indeed, biological evolution in its most general form is transcendence of the struggle for life and natural selection by entropic forms of interaction through ascending levels of symbiosis from cell organelles to biotic communities and ecosystems [17], but such evolutionary processes as entropy reduction are peculiar not only to biotic systems, but also to the biosphere as a whole.

Therefore, history enters development as heredity and is projected into the future by means of the epigenetic renovation of the developmental program. This principle determines the relationships between ontogeny and phylogeny at the genomic, organismic, and ecosystem levels, converting complexity of developmental programs into the directedness and determinism of evolutionary developments [17,18].

In evolution, the irreversibility of general progressive self-development with the cyclic nature of individual processes is important [19]. Biotic evolution is an inevitable result of maintaining the stability of the system of the highest rank relative to the organism [20]; the population is regulated by the biogeocoenosis [21] in the hierarchical structure of biotic and biocosmic systems of the biosphere.

The authors' familiarity with the problem of biospheric evolution showed that Western scientists very rarely use the literature from eastern Europe, so we intentionally used predominantly this literature in our discussions to create and maintain some informational bridges.

## 2. Fractal Structure of the Biosphere

The total Earth's ecosystem is called the ecosphere [22]; it seems that this is the real biosphere, the bioinert cover of the planet. The existence of different ecosystems at certain periods of the Earth's history is confirmed by the paleontological record. Attempts to present the biosphere as a single system closely to V.I. Vernadsky's concept [4,23] lead to an overly complex and controversial picture [24].

The biosphere is not only an area, a receptacle of living things, and not a totality of living things, but is a dialectical unity of living and non-living things interacting with each other. The living and the inert matter, possessing in many respects opposite properties [4], form a system with new properties. The concept of a bioinert system introduced by V.I. Vernadsky [23] for the biosphere is dialectical, because it unites opposites into something like a fundamentally new system, a system of a completely different type.

It is obvious that the entire totality of living matter, the living cover of the Earth (living matter [23,25], Geomerida [26,27], biocholida [28]), did not remain unchanged for billions of years. It has been evolving, and the biosphere has seen an increase in the richness of forms and the complexity of relationships. The correspondence to the habitat is an obligatory condition for the existence of living things [19]. At the same time, life is not a passive contained, and changes and development of geoholida occurred under significant pressure of biotic factors [29].

The biosphere has a fractal nature. There are the largest (biosphere) and the smallest (ecosystem, biogeocoenosis) bioinert systems on Earth. The similarity between them consists in the fact that they transform solar energy and maintain relatively closed cyclic processes of substance exchange. The fractality of the biosphere consists in this functional similarity of the parts and the whole. For this reason, it is impossible to consider the biosphere as "the largest ecosystem" [30].

Four substructures, the largest elements of the biosphere system or biospheromerons, can be distinguished in the biosphere [28]. The first one is associated with V.I. Vernadsky's ideas about the most active parts of the biosphere ("condensed life", "films"), which in the ocean are represented by the photic zone. Here, the transformation of solar energy by photosynthetics takes place. In the bottom zone, there is the transformation and accumulation

of organic and mineral substances, and the intermediate zone is functionally almost inert. One terrestrial film of life is identified on the continents.

The organizability of the biosphere is a consequence of its systematic nature. A system is a set of elements that have certain properties, namely, discreteness, hierarchy, historicity, and dynamism. The elements of a system do not exist outside of the connections between them, and a complete system never remains unchanged; it develops and evolves. At the same time, ecosystem elements can exist and reproduce not in any but only in a certain range of environmental parameters [31,32].

One of the aspects of the biosphere's organization is the broad structural and functional convergence of ecosystems. In similar conditions, biotic communities and ecosystems are formed that are similar in character, although differing in "details", species composition, and spatial features. This allowed us to identify a quite limited number of "typical", generalized biotic communities or biomes on land [33], and just a little more than 10 biogeomes [34,35]. This is the structure of the biosphere: its fractal functional elements are biospheromerons, biogeomes, and ecosystems–biogeocenoses.

### 3. Basic Conceptual Framework

The ideas, principles, and concepts on which we will base our further reasoning are the following:

#### 3.1. Principle of Existence

The principle of existence "here and now" (haesseitas): The concept of "haesseitas" was introduced into the field of evolutionary process analyses to denote the philosophical principle of the existence of reality [36]: it is what we observe here and now as really existing, real. The term itself belongs to the XIII century philosopher Duns Scott.

#### 3.2. State and Interstate

The concept of state and interstate: A given state of an object or system is represented as a time slice of all its characteristics. Each state is limited in time [37]. There are transitions or interstates between states [38]. The concept of state is closely related to the principle of haesseitas by a dialectical connection, since the very concept of "state" from the point of view of the "here and now" principle looks ambiguous: one can distinguish "instantaneous" states and states lasting. The state (here and now) can be infinitely fractionalized due to the property of time: between any two moments of time, there will always be a third one, i.e., there is no minimum segment of time [39]. In fact, the ancient philosophical constructions of Democritus and Cratylus suffered from non-dialecticism; they denied the diversity of states. We are constantly confronted with long-term, persistent conditions [40]. In biocenosis, there is change in individuals, and the composition of trophic groups changes; nevertheless, in general, we deal with the same biocenosis. As pointed out by V.A. Krassilov [17], "states" in the application to living things differ qualitatively. To understand evolution, it is necessary to grasp the idea of "non-evolution", which in the first approximation can be represented as a state of no change, however false the simplification may be. No living thing can exist in such a state, because "no change" means "no life".

The change in states builds a picture of the world not as a frozen world or unchanging state, but a world of processes [41], rather—of a morphoprocess [42].

#### 3.3. The Concept of Enclosing

The concept of enclosing and contained: The integrity and individuality of a system is determined not only by its internal properties, connections between elements, but also by the nature of interaction with the environment. The relationship between the enclosing and contained determines the hierarchical structure of bioinert systems. The dialectics of the enclosing and contained for the biosphere is that one cannot mechanically consider the geoholida as the enclosing for the bioholida—the enclosing and the contained create a new system. At the same time, it is also true that the enclosing cannot arise before the

contained [19], which implies that for the biosphere to arise, certain conditions must have existed, which were created by the geospheric cover of the Earth.

### 3.4. Principle of the Pressure of Life

The principle of the pressure of life: An important component of both Darwin's theory of evolution and the doctrine of the living matter of the biosphere is the law of unlimited growth in the number of living organisms [23,43]. This creates the effect of the pressure of life. The intensity of this pressure is a measure of the activity of living things. In addition to the quantitative aspect of the pressure of life in evolution, there is a qualitative one, which is manifested in the appearance of new and new forms of living things. If the mechanism of the appearance of new species and forms is still debated, the increasing richness of life forming in the history of the biosphere in the process of evolution is one of the undeniable empirical generalizations. The qualitative aspect of life pressure has the form of a hyperbolic curve of increasing taxa richness [44]. For most large groups of organisms, starting from the Cambrian, an ever-increasing flow of new forms and species is known. Because of this, we should probably expect a significant decrease in the growth of the richness of forms and species on evolutionary and geological time scales, similar to the logistic curve of population growth and stabilization. The qualitative pressure of life will be increasingly counteracted by the limitation of opportunities for the emergence of new ecological niches and the limited capacity of the environment on a planetary scale. In addition, it is becoming evident that the role of humans is increasing in this process of limiting the growth of species richness.

We draw attention to the fact that the concept of natural selection and the struggle for existence are far from being the only provisions of the theory of the origin of species as interpreted by C. Darwin [13,45].

However, each impact as well as counteraction are energy processes. An example from river hydraulics may be useful here. The water pressure of a stream is resisted by friction forces with the bottom, which leads to the appearance of rhythmic "clots of energy" on the bottom, as a result of which the bottom of the stream is not smooth, but is covered with periodic clusters of dunes, and metameric bottom morphological structures are formed. Also, energy preconditions of the relationship between flow pressure and bed resistance form metameric periodic structures in the form of small ponds and rolls in mountain streams [46]. Thus, there is not a progressive but a periodic process.

### 3.5. Adaptation Activity

Adaptation activity: The interaction between the enclosing and contained has a wide range of diverse properties, but the most common of them may be activity and passivity. For biotic systems, activity is essential [47]. Living systems actively rearrange their structural and functional organization, both in the aspect of modernization in accordance with the need for adaptations to the environment, and in the aspect of influence on the environment. However, any activity is associated with energy expenditure, so passive ways of adaptation are also widely used.

### 3.6. The Principle of Actualism

The principle of actualism: This principle seems to be equivalent for both the past and the future, not only as "the present is the key to understanding the past", according to C. Lyell (cited according to [48]). It can be extended: past, present, and future biospheres are connected by the unity of historical development processes based on common regularities and laws. The study of the geology of another planet (in this case, Mars [49]) would be impossible if it were not initially assumed that the basic processes of mineral formation and rock formation there and on our planet were similar.

The principle of actualism is designed to address one of the key questions that is posed when discussing the issues of development and evolution: how can we move from what we actually observe now to what was and probably will be? An analogy can be established

on the basis of a combination, a configuration of properties for which there is a space of logical possibilities. It is the principle of actualism that allows us to reconstruct past states based on configurations of properties of the present, thus reconstructing (or predicting) trends of development.

### 3.7. Information in Evolution

Information in evolution: The acquisition of information is the elimination of uncertainty, which disappears in different action [50]. The amount of information transmitted in one choice between equally probable alternatives is taken as the unit of its quantity. Life as a process is associated with entropy reduction, which corresponds to the active accumulation of information [36]. The correspondence of a given state of the system to the conditions of existence as a choice/selection from several possible states is an informational act [51]. Transition from one state to another is based on information transfer. In this case, as follows from the principles of cybernetics, in particular, as applied to biotic processes [21], information transfer is accompanied by noise, which acts as a source of novelty. Not only the material body, the object, but also information about it, about its structure and system properties, are transferred to the subsequent state of the system.

### 3.8. The Concept of Selection

The concept of selection: Selection in evolutionary processes should be considered much broader than "Darwinian" selection [43] or "survival of the most successful" in the struggle for existence. Supplementing the "struggle" itself with cooperative relations, the advantages offered by symbiosis do not change the principles of "Darwinian" selection. This is a particular "biological case". Selection in the broad sense is the preservation and existence of the contained, corresponding to the conditions of the enclosing. The transition of a system to a given state, and the preservation of itself, is an act of selection associated with the reduction in other possible states.

The act of selecting a state is an act of removing uncertainty, i.e., it is an informational act, and the transition from one state to another is associated with the transfer of information.

### 3.9. The Principle of Emergency

The principle of emergency: An emergency, as a property of systems, consists in the fact that the laws and principles of the system as a whole are not applicable to subsystems and elements. Many ecosystems and types of ecosystems in the process of evolution behave rather chaotically in relation to the biosphere system. The organization of a hierarchical system is an ordering at higher levels of chaos at lower levels. Probably, a certain amount of chaos of elements is an important prerequisite for the vital activity and, moreover, for the development of the whole system; at the same time, the existence of certain trends with a certain direction in the evolution of the integral biosphere is undoubted. Chaotic nature is a source of novelty, which is expressed in information noise.

## 4. Dialectical Principles and Categories

### 4.1. Continuity and Discreteness

Continuity and discreteness: The evolution of the biosphere was and is continuous, without gaps, due to the absence of multiple acts of the origin of life. At the same time, in the history of the biosphere, there were long states of relative stability, and small changeability of essentially important characteristics, to the extent that properties within the state are indistinguishable. This is a continuum of the first kind [52]. In the case of relative discreteness of lasting states separated by interstates, changes occur in the unified continuum of the evolutionary system, which determine a continuum of the gradient character or a continuum of the second kind: the subsequent state differs from the previous one, the initial states are very different from the present ones, and we are dealing with the same continuum evolutionary system.

If we consider punctuated evolution [53] as an alternative to a purely gradualist approach, then we cannot consider it as an evolutionary discontinuity of the biosphere as a whole, including the biocholida.

The unity of the whole evolutionary system of the biosphere determines a certain fundamental similarity of successive lasting states (due to additivity and continuity), which leads to a new form of continuum—a metameric continuum (continuum of the third kind). As an illustrative model, we can cite a cascade of reservoirs on a river [54] or a chain of ponds that are separated by sections of the river.

## 4.2. Reproducibility and Uniqueness

Reproducibility and uniqueness: Reproducibility is the phenomenon of the emergence of new objects that are part of a group, a class of already existing, or previously existed. Relating them to one class presupposes the presence of similar qualities and properties, so dialectically, reproduction is opposed to uniqueness, individuality, and inimitability. The phenomenon of reproducibility is widespread both in cosmic and living nature, and in human activity [55]. The normal functioning of systems is based on the balance of interaction between reproducibility of similar elements and uniqueness of unique elements in their structure. No system can exist with the identity of its elements, which can be endlessly reproduced, as well as with their absolute individuality and inimitability, because the functioning of the system is based on a diverse interaction of both homogeneous and different elements. Reproducibility participates in the processes of maintaining a certain level of diversity necessary for the system. It is obvious that evolution as a process of creating and fixing the new is a process of breaking absolute reproducibility as an exact replication of the past state, and past form and function. Here, apparently, it is necessary to take into account temporal and spatial scales. Is the birth and development of stars [8] the reproducibility of cosmic objects or should the very process of their development be considered as evolution?

## 4.3. Succession and Novelty

Succession and novelty: These are dialectically related concepts that define conservative and revolutionary principles. In this case, the subsequent state always contains both the "inheritance" of the past and a certain amount of novelty. Obviously, the loss of the old, changing the structure of the system and the configuration of elements, as a consequence, also leads to the emergence of novelty. Too large "portions" of novelty can significantly weaken the stability of the system [56]. The acquisition of novelty occurs during the interstate period; for the new state, it becomes "routine". The level of acceptable and sufficient novelty is set by selection based on the suitability of the enclosing to the contained. Novelty in reproducibility processes is information noise, which is the material for selection. The emergence of new configurations in a system [8] can be considered as novelty for the system, but not at the level of a set of elements, since new configurations can be created from non-new elements. This should probably be called reproducibility variability. Islands close to the mainland are inhabited by the same species and form the same ecosystems as on the mainland. On significantly distant islands, an intensive process of species and form formation can take place, resulting in significant novelty in both the composition of elements and their "ecological configurations" [57].

Inert systems in their development are essentially limited by the laws of physical processes and chemical interactions; biotic systems are much more "free" in creating novelty, although they also exist in the "corridor of logical possibilities" [19].

Novelty in evolution is created rather than formed from already existing variants [17]. Creativity becomes crucial in the evolutionary process. Henri Bergson developed a holistic concept of creative evolution [58], considering biological evolution as arising from the systemic organization of life. Evolution is a teleological process of internal causality, but unlike the teleology of a human, the individual, finality in nature is a phenomenon of all life in the biosphere, encompassing the entire population of the planet.

### 4.4. Coherence and Dissociation of Processes

Coherence and dissociation of processes: Coherence (from Latin cohaerentio—connection) of the system elements is determined by "the prohibition of development of the absolute whole" [56], in which case everything or nothing can be changed at once. The elements of the system always have a certain individuality and independence. The presence of connections in ecological systems is considered within the framework of the law of ecological correlation [24]: all living and inert components of the system correspond to each other. Nevertheless, the links cannot be absolutely rigid. That is why V.A. Krassilov [15] introduces the concept of coherent evolution, i.e., not static interrelationships, but interrelated changes. Indeed, if, as V.A. Krassilov gives an example, Mesozoic vegetation has not undergone significant changes for many millions of years, it does not mean that there were no changes, both ecological and evolutionary, at all, but that they occurred coherently, which maintained the stability of the biosphere system as a whole. The incoherence of crisis phases, interstates between stable lasting states, is precisely because stable, interconnected ecological processes are disturbed.

### 4.5. Stability and Variability

Stability and variability represent one of the most important dialectically related pairs of concepts in evolutionism. As is well known, it was variability that was considered in Darwinism as one of the prerequisites for evolution. Variability creates noise in the transmission of information. It is important to note that the stability of the organization of systems is possible with the existence of a more or less complex regulatory apparatus that protects the normal creation of forms from possible disturbances by random deviations of environmental factors [59]. It follows from this that maintaining the stability of systems is an energy-consuming process. But, at the same time, invariability and stability of the whole can be achieved only by changing parts [20]. As it was noted [19], the greater the complexity of a system, the less its stability. The author draws this conclusion from the fact that periods of global restructuring occurred more and more frequently on the geologic time scale.

### 4.6. Additivity and Reduction

Additivity and reduction: The evolution of the biosphere has an additive character, i.e., there is an accumulation of certainty of its "achievements". The prokaryotic hydrospheric biosphere became a part of the intervening for its subsequent states, and "plantium" or the appearance of terrestrial macrophytes became a part of the containing for the subsequent states of the epigean part of the biosphere and the biosphere as a whole. Novelty is superimposed on the already existing. The notion that evolution is solely a composite of novel things is one of the major misconceptions about the development of a biospheric system. The dialectical opposite of additivity is a decrease, reduction, which is connected with selection. The dialectic of these opposites is that accumulation cannot be infinite and must be limited by loss, freeing up a "place" for novelty, maintaining the cyclicality of processes.

## 5. Concept of the Biosphere Evolution Trend

A trend can be regarded as an information channel through which information is transferred from state to state in an evolving system, which strictly concerns one property of the system. A system has many properties, a developing system has many trends, and their time slice creates the structure of the system state. R. Margalef [51] distinguishes three such information channels in time: genetic, ecological, and ethological or cultural–social. In any evolving system, diverse and characteristic channels–trends can be distinguished. As for the evolution of a biospheric system, we should distinguish abiotic or environmental, biochemical, physiological, taxonomic, ecomorphic, symbiotic, ecosystem, ethological–social, and anthropo-cultural trends. In each state, the "power" of the trend is different,

which creates, in fact, a "portrait" of the structure of the state of the developing system of the biosphere.

## 6. Organizability of the Biosphere as a Morphoprocess

Considering the principles of organization of biotic and bioinert systems, V.N. Beklemishev [42] uses the concept of a morphoprocess. Although he applied this concept for an organism, it is much broader and can be used for different self-regulating systems, including the biosphere [60]. A morphoprocess is the maintenance of a relatively stable form of the whole due to constant change and renewal of parts. It is the dialectical unity of the relative stability of the state of a given form and the constantly changing process that maintains the form, and the morphoprocess supports the continuity of space–time of the biosphere.

## 7. Evolutionary System of the Biosphere

The existence of any system in time is a change in states. In a system that develops/evolves, the change in states occurs with their increasing complexity. For example, the system of organism development, ontogenesis, can be represented as a series of states: juvenile age, maturation, reproductive state, maturity, aging. The main trends uniting ontogenesis states are physiological, somatic, ecomorphic, ethological, and social. The temporal slice of states is a slice of the totality of trends at a given time (Figure 1). In the succession of ecosystems, as in ontogenesis, there is a definite change from one state to another. As E. Odum believed, they are united by quite a large number of trends: energetic, structural, symbiotic, life strategies, etc. [61]. However, both ontogenesis, as the development of an individual, and succession, as the development of an ecosystem, are quite significantly determined. The first process is determined by the genetic program, and the second by a certain set of available populations and characteristic climatic conditions (Figure 1).

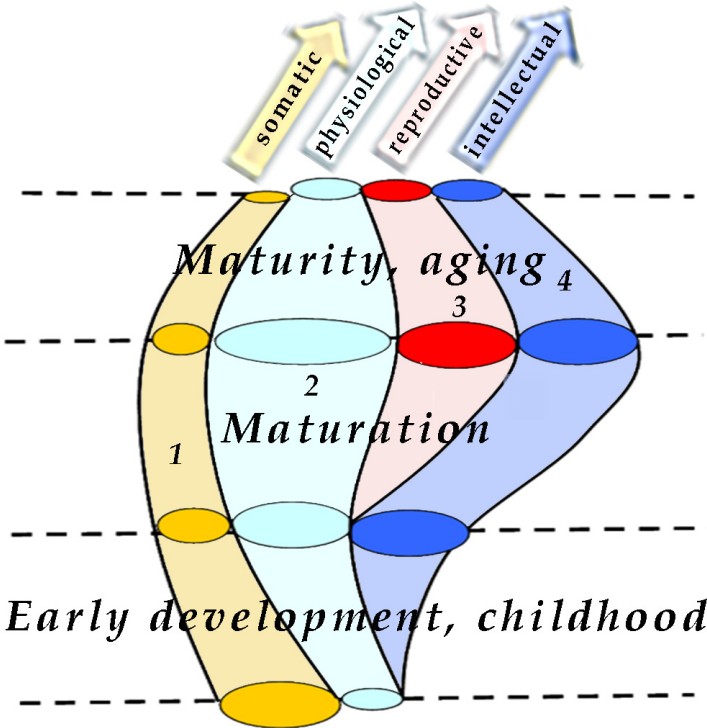

**Figure 1.** Ontogenesis as a metameric system: (**1**) somatic trend; (**2**) physiological trend; (**3**) reproductive trend; (**4**) intellectual trend. Dashed lines separate the three states of the system—states of the ontogeny system. Adapted from [16].

As far back as in the XIX century, geologists and paleontologists discovered a rather clear separation of essentially homogeneous stratigraphic elements of geological systems.

The change in the composition and abundance of biotic fossils in successive layers of sedimentary strata is the only direct evidence of the reality of evolution and the main source of ideas about unidirectional geologic time [62].

Such a complex system as the biosphere has a huge number of elements characterizing it at a given moment, and in a given epoch or period of its existence and development, but it is possible to identify a small number of basic trends, the slices of which, or rather the structure of slices, determine the character of each state (Figure 2).

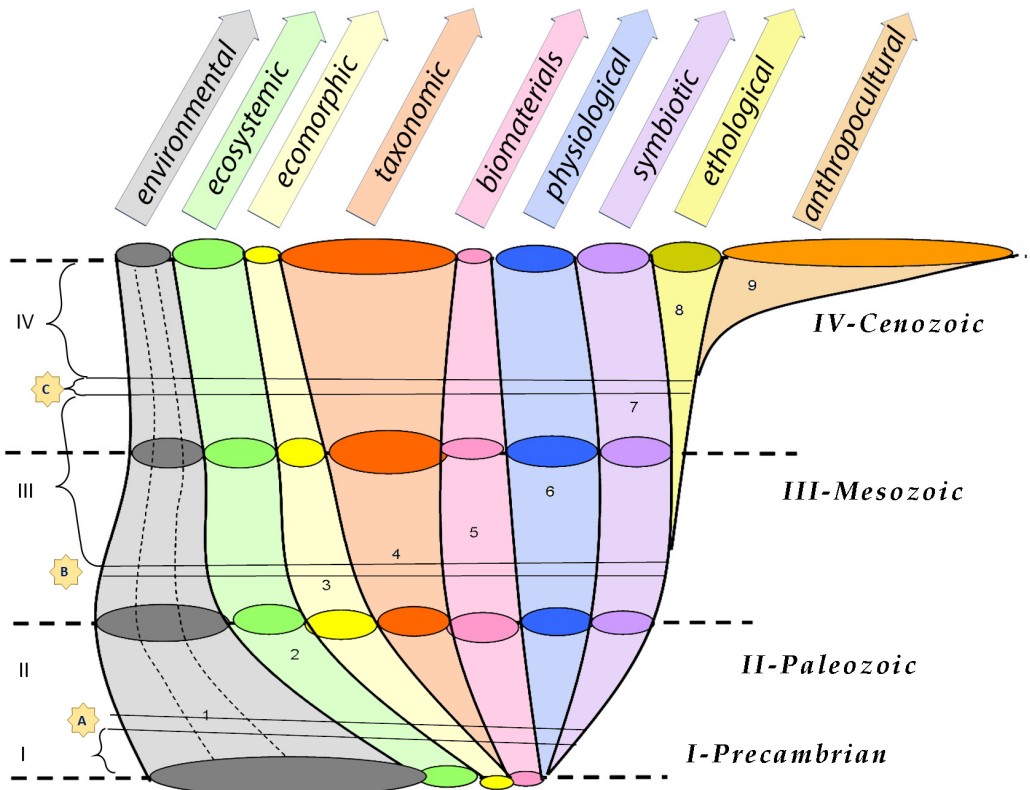

**Figure 2.** Trends of the evolutionary system: (**1**) abiotic/environmental/geoholida; (**2**) ecological/ecosystemic; (**3**) ecomorphic/life forms; (**4**) taxonomic/biodiversity; (**5**) biochemical/biomaterials; (**6**) physiological/metabolic processes/adaptations; (**7**) symbiotic/cooperation/consortiogenesis; (**8**) ethological/social; (**9**) anthropo-cultural; (**I–IV**) states of the evolutionary system; (**A–C**) inter-states. Adapted from [16].

In the group of abiotic trends, we should single out the trends of evolution of the lithosphere, hydrosphere, and atmosphere, i.e., those cosmic elements of the biosphere that create biotopes as elements of ecosystems or a unified inert "skeleton" for the living part of the biosphere. Regarding a single world ocean and periodic unity of continents, when they are separated—the existence of large continents, and a single system of air mass movement in the atmosphere—all these were the most important prerequisites for the formation of a unified structure of the biosphere.

### 7.1. Biotic Trends Are Diverse

Biotic trends are diverse: The most "popular" in evolutionary studies is the taxonomic trend. In fact, it is the trend of genetic evolution. The appearance of new species and forms is a historical, paleontological fact. It is important for the evolution of the biosphere that this trend becomes more and more "powerful".

An organism exists in nature not only as a carrier of genetic information, and belongs not only to a certain species, but also resides in the form of a specific real ecomorph; therefore, one of the important evolutionary trends is the ecomorphic trend, the trend of

life forms. There is both replication, reproducibility of different ecomorphs, life forms, and their emergence in the process of evolution.

### 7.2. Cooperative Symbiotic Relationships

Cooperative symbiotic relationships in the broadest sense are strong in nature. They have become more complex, passing through certain stages of evolution. Symbiotic relations were important not only in purely biotic systems, but also changed the relations of living things with the cosmic environment. Therefore, the symbiotic trend can also be considered as one of the most important in the evolution of the biosphere.

The biochemical evolutionary trend is the history of formation of both new biological "material" and biochemical processes. The process of creation of chemical bases of the regulation of organismal processes as well as chemical mechanisms of the regulation of relations (information transfer) in various associations of organisms was also important.

The issue of ecosystem evolution has been discussed many times [17,29,63]. The validity of singling out such an evolutionary trend is obvious, since it is ecosystems that represent the units of its structure. The evolution of ecosystems cannot be adapted to Darwinian principles. There are no random mutations, competition, and struggle for existence in the Darwinian sense. There is a process of biotic self-assembly, community-level adaptations, and succession. The evolution of ecosystems consisted in successive changes in the nature of matter cycles, and taxonomic and ecomorphic structures of biotic communities. Changes in the ways of obtaining and transforming energy should be added to this. The limited types of ecosystems, biogeomes, indicate the predominance of convergent processes in this area of biosphere evolution. Actually, the evolution of ecosystems is the evolution of biogeomes, the coherent evolution of many ecosystems towards the formation of their unity.

### 7.3. The Anthropo-Social-Cultural Trend

The anthropo-social-cultural trend has existed since the beginning of the evolutionary period of the appearance of direct biological ancestors of humans on Earth; it is based on close ties with the general biological social trend. But, as noted by V.A. Krassilov [13,64], during the past 30–40 thousand years, humans as a biological species have been in a state of morphological stasis—in the evolutionary sense, a human is unique, as their evolution is almost completely shifted to the field of culture and technology. Thus, the social trend of the biosphere evolution has transformed into a socio-cultural trend, and the development of human society and culture is a continuation of its evolutionary biological trend [17].

## 8. Evolutionary System and Metameric Model of Biosphere Evolution

The state of the biosphere in certain periods is connected by trends, which form an evolutionary system. Lasting states, when the system remains self-identical for a long period of time, are replaced by new states through interstates. Thus, a peculiar type of continuum, the metameric continuum, is formed. Although the states are different, they remain states of the same evolving system. All the same biospheric functions are performed, details change, and the additive process is going on, but the integrity, the continuum of the global evolutionary process, is preserved.

The comparison proposed for river ecosystems by the ecologist and morphologist V.N. Beklemishev [65] can be illustrative and useful here. Their ecosystems have a metameric structure, i.e., the alternation of rolls and swims. This is one of the main differences between a natural watercourse, a river, and an artificial channel, a property of a self-sustaining lotic ecosystem. Regarding the "metameric" evolutionary process of the biosphere, comparing it to a riverbed is not only a beautiful metaphor, but it reflects, obviously, certain regularities. First of all, let us pay attention to the geological and paleontological chronicle of the biosphere. It is divided into eons, epochs, precisely because small boundary time periods, when global extinctions of former forms occurred and new ones appeared, are separated by longer periods of relatively calm "coherent" development.

Thus, in the evolution of the biosphere, we deal with three types of continuity. Continuity of the first kind can be observed when there is a continuum of indistinguishability within a lasting state: during a long geological time, there are no cardinal changes in the structure of the biosphere elements and connections between them. At the same time, the development of an integral biosphere system occurs in a continuum of the second kind: in the case of continuity of biosphere life, the differences between the earliest and subsequent states are very significant and obvious. Moreover, these differences are connected to complication and development, which is, in fact, evolution. Finally, there is a continuum of the third kind—metameric—with a regular alternation of states lasting for a long time and relatively short interstates. The complex system of continuums is the most important feature of the biosphere's organization in space–time.

Thus, all the above-mentioned principles play their role in the organization of the evolutionary system, but several key ones can be singled out:

- The succession of life, with constant change in its content, from the structure of the organism to the structure of biotic communities;
- Additivity, the accumulation of fundamental acquisitions of the evolutionary process, the creation of novelty, and its consolidation in the structures and functions of the biosphere on a hierarchical basis;
- The pulsation character of the evolutionary process is determined by the interactions between the pressure of life and the pressure of the environment, and the contradiction between the unlimited growth of living things and the limited capacity of the environment;
- Maintaining the biosphere's organization and evolution requires energy inputs.

The history of the biosphere looks like a metameric picture of the change in lasting states, delimited by interstates, on the geologic time scale. The change in states represents a rupture of the continuum of the first kind. The pressure of life and the pressure of the environment are in a certain equilibrium in the states of coherent evolution, but the pressure of life as an active component of the system leads to a violation of the balance; there is a crisis, extinctions, the redistribution of ecological niches, and then a new cycle begins. Additivity leads to the fact that in each cycle, the balance is established at a higher and higher level.

## 9. Past and Future

The conditions with which life is compatible have limitations on planet Earth both in the past and in the future [66]. Regarding the cessation of the biosphere's existence, as well as its emergence, emergence is possible only as a whole, the whole system. In this regard, it is impossible to preserve humanity outside/after the existence of an effectively functioning biosphere. As V. I. Vernadsky [25] believed, the biosphere could only emerge immediately as an integral system, at least in the simplest variants of energy transformation, exchange formation, and the substance cycle. The scenario of the "origin of life in the form of organisms" cannot be considered as the only possible one. That is why the author of [51] hypothesized the emergence of the biosphere in the form of protoecosystems in which life existed in pre-organismal forms. The self-reproduction of organisms subsequently became one of the key "achievements" of early evolution. Based on strictly empirical data, V. Vernadsky concluded in the early twentieth century that non-organismal life did not exist on Earth, and there is no confirmation of this now. In this case, the most probable way of the origin of the biosphere could be associated only with the hypothesis of panspermia, the transfer of life in water organisms between cosmic bodies, on which the conditions corresponded to what is necessary for existence.

But even this hypothesis has no empirical confirmation because transport with subsequent development could occur only in pools of "fragments of biospheres". The transport of organisms in the cryptic state contradicts the idea of the very existence of life only within the biosphere. In such a case, should we not pay attention to the universal phenomenon of reproducibility? If new planets and new galaxies appeared, why can't life reproduce

everywhere where conditions correspond to its appearance? With the huge diversity of conditions in space, life on a cosmic scale must also have a huge diversity. This is the highest level of biodiversity. In this case, we come, following R. Margalef [51], J. Bernal, and M. Kamshilov [67], to the hypothesis of the existence of life before the appearance of living organisms, at least in such a form as we know them now.

In the history of the biosphere, there have been many crises of different scales; they led to a change in states and were relatively brief, which allows us to call them evolutionary episodes [64]. However, there has never existed a phenomenon of global influence on the whole system of the biosphere of one biological species, as it happened with humans. B. Vernadsky believed that this influence is directly related to the formation of a sphere that was in mind—noosphere—a new state of the biosphere [4]. The systemic theory of evolution [13,64] considers the development of human civilization as a direct continuation of the processes that preceded the emergence of humans. The human population has the same principles of organization as the rest of living beings; the number was regulated by predators, diseases, parasites, competing species, and even cannibalism and infanticide, but mainly by the quantity of food resources. Adaptations at the first initial stages of human evolution were similar to other organisms: immunity enhancement, competitive struggle, migrations in search of rich food resources. But the peculiarity of humans was the formation of new types of adaptations: the creation of tools and technologies, and then the creation of agro-ecosystems, and later urban ecosystems and techno-ecosystems.

If we return to the fractal structure of the biosphere as a complex system, which is based on its smallest part—the ecosystem—it should be noted that humans have created and continue to create new types of ecosystems through their activity and labor. Here, we should emphasize the duality, "synthetic nature" of anthropoecosystems, rather than the existence of the technosphere [68], as some superstructure over the biosphere. The "technosphere" is an anthropogenic part of the geoholida. Since the inhabited space of the planet is of course limited, anthropogenic ecosystems will inevitably exert pressure on natural ones. The modern ecological crisis, among other things, consists in the fact that anthropogenic ecosystems, unlike natural ecosystems, need a large amount of additional energy, which is supplied by humans [69]. One of the most important tasks of human–nature interaction is to find harmony, primarily energy harmony, between natural and anthropogenic ecosystems.

## 10. Conclusions

The evolution of the biosphere must be considered systematically. Its fractal structure exists due to the emergent properties of each level. The hierarchy of the system is supported by cybernetic control and interrelationships between the intervening and the interfering. The evolution of the biosphere occurs in a specific space–time, where different types of continuity determine the relationships between extant states and developmental trends, forming an evolutionary system.

The organization of the biosphere has many manifestations. The huge "biodiversity" is minimized to a relatively small number of functional groups, ecomorphs. The diversity of ecosystems is rolled up into a small number of biogeomes, and an even smaller number of biospheromerons. This blockiness is fractal in nature. Metameric evolution is a morphoprocess: a constant renewal on the scale of its geologic biosphere time, and a vortex of life.

**Author Contributions:** Conceptualization, A.P. and S.B.; methodology, A.P.; validation, A.P. and S.B.; formal analysis, A.P.; writing—original draft preparation, A.P. and S.B.; writing—review and editing, A.P. and S.B.; visualization, A.P.; supervision, A.P. All authors have read and agreed to the published version of the manuscript.

**Funding:** This research received no external funding.

**Acknowledgments:** This is a translation and further development of the ideas in the publication "The evolutionary system of the biosphere: a dialectical approach" originally published in Ukrainian

by the National Museum of Natural History, NAS of Ukraine, *Geo & Bio* **2023**, *25*, 32–50. [16]. The translation and further expansion were prepared by Protasov, A.A. and Barinova S.S. Permission to use parts from the publication in Ukrainian was granted by the National Museum of Natural History, NAS of Ukraine, *Geo & Bio* journal, copyright holder Protasov, A.A., and/or author's name Protasov, A.A. The authors would like to express their gratitude and appreciation to the Editorial Board of the *Geo & Bio* journal for permission to use materials and some text fragments and figures of the previously published article for this paper: Protasov, A.A. The evolutionary system of the biosphere: a dialectical approach. *Geo & Bio* **2023**, *25*, 32–50. (In Ukrainian)— reference [16]. The authors are also grateful for the partial support of the Israeli Ministry of Aliyah and Integration.

**Conflicts of Interest:** The authors declare no conflicts of interest.

## Glossary

**Biosphere**: The system of all ecosystems on the planet that interact with each other to a greater or lesser extent at a given period of time; **Biogeome**: A set of ecosystems that are similar in their structural and functional organization (e.g., rainforest biogeome, desert biogeome, river ecosystem biogeome, or rheobiogeome); **Biome**: The biotic part of a biogeome, and the totality of all its biocoenoses; **Geome**: The abiotic part of a biogeome, and the totality of conditions, environmental factors, and resources that support life; **Biospheromeron**: The largest element in the structure of the biosphere. There are four of them: a surface photic and productive ocean biospheromeron, a dysphotic bottom destructive biospheromeron of the ocean, the intermediate inert zone between first and second biospheromerons, and a surface biospheromeron of continents; **Biocholida**: The totality of all living things in the biosphere; **Geoholida**: The totality of all abiotic conditions, factors, and resources that support life in the biosphere; **Living matter** (in Vernadsky's understanding): The totality of all living organisms of the biosphere in the first approximation similar to Geomerida and Biocholida.

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
