# Peer review of "The Evolutionary System of the Biosphere and the Metameric Concept of Its Evolution: From the Past to the Future"

_encyclopedia, doi:10.3390/encyclopedia4020058_

Round 1
Reviewer 1 Report
Comments and Suggestions for Authors
The concept of the evolution of the biosphere and its structural and functional elements - biogeomes – proposed by the authors is highly original and new. The concept presented in the article develops the theoretical views of one of the authors – A.A. Protasov on the process of evolution of the biosphere, which he had previously published separately in journal articles and monographs. The key concept – biogeome, as it seems to me, is largely analogous to the concept of "biogeocenosis" proposed by V.N. Sukachev. The authors introduced the terms – geoholide as a set of biotopes suitable for life, and also bioholide - a biosphere set of biocenoses, therefore, similarly organized and functioning ecosystems, by analogy with biogeocenosis, are united by them into a biogeome as the first highest hierarchical level of organization of the biosphere. The authors of the article proposed other new terms, such as biospheromeron, i.e. part or fragment of the biosphere and justify the associated metameric aspect of considering the evolution of the biosphere.
One of the important parts of the proposed concept is partly borrowed from V.N. Beklemishev’s idea of a coherent morphoprocess, which the authors extend to all elements of the biota. However, I should note that the use of the concept of ecosystem, and not biogeocenosis, as the lower level of the hierarchy of biosphere structural elements, in my opinion, seems unsuccessful, since the ecosystem does not correlate with the spatial chorological division of the biosphere into biogeomes, since it does not reflect the spatial-structural aspect , but exclusively functional (the entire biosphere, in a certain sense, is a macroecosystem). An ecosystem is a relatively autonomous functional self-organized system of flows of matter and energy, and a biogeocenosis, while maintaining these properties, is localized as a chorological unit. In this case, the need arises to construct parallel biogeomic and ecosystem hierarchies of the biosphere. However, the authors offer a mixed version, relying on the one hand on the ideas of V.I. Vernadsky about the concentration of living matter (implying functioning ecosystems), and on the other hand, on the views of V.N. Beklemishev about the hierarchy of sets of similar biocenoses. Somewhat left out of the discussion were the ideas of V.V. Zherikhin, to whose article the authors referred, that the evolution of communities and biomes can sometimes occur without the evolution of species – phylocenogenesis can occur without phylogenesis. The evolution of communities can be carried out in the form of a functional change in the composition of communities without a significant change in the morphoprocess of specific species, i.e. without speciation. Therefore, the authors should probably explain how phylocenogenesis and phylogenesis are related in their version of the evolution of the biosphere?
There are minor typos in the text of the article that need to be corrected. So in line 127 in the bibliographic reference you need to write [35] instead of [335]. I propose to replace the word “interstate” with “intermediate state”. The salvo link to publications in line 53, which includes 8 cited articles, in my opinion, should be replaced by 2-3 sentences with links containing explanations and specifications of individual points by which the articles were grouped in the link.
As a result, it should be concluded that, despite some inconsistency in the authors’ ideas, I consider the concept they proposed to be a completely rational and useful solution that awakens theoretical and philosophical thought and can serve as the basis for further creative development of ideas about evolution and the structural and functional organization of the Biosphere. I believe that the article will arouse considerable interest and attract the attention of many specialists in various scientific specialties, so it should definitely be published in the Encyclopedia journal after making minor corrections.
Author Response
Dear Editor,
Thank you and the Reviewer 1 for comments.
Please consider responses to the Reviewer 1 comments below.
With best regards,
Prof Sophia Barinova,
Corresponding author
Reviewer 1 Report 1
Comments and Suggestions for Authors
Reviewer: The concept of the evolution of the biosphere and its structural and functional elements - biogeomes – proposed by the authors is highly original and new. The concept presented in the article develops the theoretical views of one of the authors – A.A. Protasov on the process of evolution of the biosphere, which he had previously published separately in journal articles and monographs. The key concept – biogeome, as it seems to me, is largely analogous to the concept of "biogeocenosis" proposed by V.N. Sukachev. The authors introduced the terms – geoholide as a set of biotopes suitable for life, and also bioholide - a biosphere set of biocenoses, therefore, similarly organized and functioning ecosystems, by analogy with biogeocenosis, are united by them into a biogeome as the first highest hierarchical level of organization of the biosphere. The authors of the article proposed other new terms, such as biospheromeron, i.e. part or fragment of the biosphere and justify the associated metameric aspect of considering the evolution of the biosphere.
RESPONSE:
It should be noted that nowadays, especially in the Western, English-language literature, the key concept of ecosystem has almost completely acquired the meaning as interpreted by E. Odum, i.e. in the functional aspect more than in the chorological one. However, the dialectical approach to defining the structure of the biosphere certainly requires paying attention to and using in cognitive terms and contrasting and comparing the two approaches - both functional and chorological. The elements of the biosphere, as quite material objects, occupy a certain position in space, on the one hand, and, on the other hand, only the interaction of elements, their functioning supports the very existence of the biosphere as a whole.
Strictly speaking, the established "tradition" of using the two concepts of ecosystem and biogeocoenosis is definitely beyond the scope of the dialectical approach. Perhaps it would be correct to introduce a new synthetic term, but upon closer examination it should be noted that both A. Tansley and V. Sukachev, the authors of these two terms, practically meant the same natural systems. This question requires further deep philosophical research.
We set ourselves more modest, simple goals. Understanding the difficulties in terminology, in this paper we propose to use the concept of ecosystem to designate the minimal unit of the hierarchical structure of the biosphere, assuming that it is related both to functional processes that support the existence of the biosphere at all levels of organization and represent real spatial formations, chorological parts of the biosphere. We also propose an approach to the structure of the biosphere as a fractal system consisting of functionally similar elements, one of the levels of such hierarchy being the biogeome as a set of similar ecosystems (biogeocenoses). Therefore, we cannot consider the biogeosome as an analog of biogeocoenosis; it is a higher-order element of the biosphere.
Reviewer: One of the important parts of the proposed concept is partly borrowed from V.N. Beklemishev’s idea of a coherent morphoprocess, which the authors extend to all elements of the biota. However, I should note that the use of the concept of ecosystem, and not biogeocenosis, as the lower level of the hierarchy of biosphere structural elements, in my opinion, seems unsuccessful, since the ecosystem does not correlate with the spatial chorological division of the biosphere into biogeomes, since it does not reflect the spatial-structural aspect , but exclusively functional (the entire biosphere, in a certain sense, is a macroecosystem). An ecosystem is a relatively autonomous functional self-organized system of flows of matter and energy, and a biogeocenosis, while maintaining these properties, is localized as a chorological unit. In this case, the need arises to construct parallel biogeomic and ecosystem hierarchies of the biosphere. However, the authors offer a mixed version, relying on the one hand on the ideas of V.I. Vernadsky about the concentration of living matter (implying functioning ecosystems), and on the other hand, on the views of V.N. Beklemishev about the hierarchy of sets of similar biocenoses. Somewhat left out of the discussion were the ideas of V.V. Zherikhin, to whose article the authors referred, that the evolution of communities and biomes can sometimes occur without the evolution of species – phylocenogenesis can occur without phylogenesis. The evolution of communities can be carried out in the form of a functional change in the composition of communities without a significant change in the morphoprocess of specific species, i.e. without speciation. Therefore, the authors should probably explain how phylocenogenesis and phylogenesis are related in their version of the evolution of the biosphere?
RESPONSE:
We are very pleased that our modest work has aroused in the respected Reviewer questions that touch upon the foundations of ecological concepts and their philosophical interpretation. We note that we specifically stipulate the issue of impossibility to consider the biosphere as "the largest ecosystem", and introduce the concept of functional fractality for the hierarchical fractal structure of the biosphere.
We are extremely grateful to the respected Reviewer for pointing out that V. Zherikhin's article addresses the issue of (in fact) non-coherence of phylocoenogenesis and phylogenesis. In a certain sense, this is also indicated by our trend approach to the development of the model of the evolutionary system of the biosphere. After all, we consider ecosystem (Fig. 2) and taxonomic (biodiversity, phylogenetic) trends as relatively isolated, self-developing ones.
Reviewer: As a result, it should be concluded that, despite some inconsistency in the authors’ ideas, I consider the concept they proposed to be a completely rational and useful solution that awakens theoretical and philosophical thought and can serve as the basis for further creative development of ideas about evolution and the structural and functional organization of the Biosphere. I believe that the article will arouse considerable interest and attract the attention of many specialists in various scientific specialties, so it should definitely be published in the Encyclopedia journal after making minor corrections.
RESPONSE:
We are extremely grateful to the distinguished Reviewer for the positive evaluation of our work
Reviewer: There are minor typos in the text of the article that need to be corrected. So in line 127 in the bibliographic reference you need to write [35] instead of [335]. I propose to replace the word “interstate” with “intermediate state”. The salvo link to publications in line 53, which includes 8 cited articles, in my opinion, should be replaced by 2-3 sentences with links containing explanations and specifications of individual points by which the articles were grouped in the link.
RESPONSE:
Necessary changes have been made to the text

Reviewer 2 Report
Comments and Suggestions for Authors
Dear authors
I think it is an interesting manuscript, but it has several problems that need to be improved. First, more than 90% of the literature cited is in Russian and cannot be verified by researchers from the rest of the world. Second, the literature cited in Russian and English is very old and should be updated (see suggestions). Third, due to the literature cited in Russian, many unknown concepts are used, which need to be known to understand the proposed hypotheses. Therefore, it is suggested that the manuscript should include a glossary (i.e., biogeomes, biospheromerons, biogeocoenosis, Geomerida, biocholida). Fourth, the figures used are not easy to follow and the legends are not very explanatory. It is necessary to read everything to understand them. The figures should speak for themselves with the help of the legends. Fifth, the history of the biosphere should be explained concerning the geological scale. Sixth, explain Vernadsky's ideas since the literature cited is not accessible.
Details:
The concept of no change should be understood as stasis (see Eldredge & Gould 1972).
The principle of the pressure of life must consider extinctions. The diversity of life is not static nor does it have growth with some trends due to large extinctions (see Futuyma & Kirkpatrick 2017). Diversification = speciation-extinction.
Improving the selection principle, the struggle of the fittest is not all that Darwin proposed (see Futuyma & Kirkpatrick 2017).
The continuity principle must consider mass extinctions. Life on Earth is not continuous because of mass extinctions.
Change taxonomic to phylogenetic. Taxonomy is a human activity of classification that is not an evolutionary property, but phylogeny of lineages is.
The ontogenetic stages of systems I,II,III,IV must be explained in terms of the geological scale. When they occur, when they begin, and when they end (Fig. 1, 2).
Finally, I think it is a very good review of ideas unknown to student readers and researchers in the rest of the world and can create a bridge between world knowledge.
Recommended Literature
Futuyma, J. D., & Kirkpatrick, M. (2017). Evolution (4th Ed.). sinauer, Sunderland, MA: Sinauer Associates.
Eldredge, N., & Gould, S. J. (1972). Punctuated equilibria: an alternative to phyletic gradualism. Models in paleobiology, 82, 115.
Author Response
Dear Editor,
Thank you and the Reviewer 2 for comments.
Please consider responses to the Reviewer 2 comments below.
With best regards,
Prof Sophia Barinova,
Corresponding author
Reviewer 2 report 1
Comments and Suggestions for Authors
Reviewer: Dear authors
I think it is an interesting manuscript, but it has several problems that need to be improved.
RESPONSE: The authors sincerely thank the respected Reviewer for the overall assessment of the work as "Interesting". This cannot but please!
Reviewer: First, more than 90% of the literature cited is in Russian and cannot be verified by researchers from the rest of the world.
Second, the literature cited in Russian and English is very old and should be updated (see suggestions).
RESPONSE: Indeed, at the very beginning of the article we specifically note that it was one of our tasks to familiarize Western scientists with the scientific heritage of the experts who wrote and thought in Cyrillic symbols. Unfortunately, this is a very bad, pernicious tendency of modern education - to think that all the most intelligent and valuable things were created exclusively yesterday, and all those who worked yesterday stand firmly and automatically "on the shoulders of giants". Probably a lot of people think that Lovelock's concept is more fundamental than Vernadsky's (if they know about the latter name) only because it, this concept, is 50 years younger. If such a paradigm had existed in ancient Rome, we might not know about Platon and Heraclitus now.
On the other hand, we are aware that Eastern European scientists are not very familiar with the development of evolutionary thought in the West, and there were and still are too many barriers to the exchange of ideas and information. Therefore, we consider this article as a call for cooperation, and we also address this proposal to you, dear Reviewer.
Reviewer: Third, due to the literature cited in Russian, many unknown concepts are used, which need to be known to understand the proposed hypotheses. Therefore, it is suggested that the manuscript should include a glossary (i.e., biogeomes, biospheromerons, biogeocoenosis, Geomerida, biocholida).
RESPONSE: This is a real comment from a real reader, who feels the lack of explanation, and here we need only thank him for his advice and create such a glossary, although references are provided throughout the text:
Glossary
Biosphere: The system of all ecosystems on the planet that interact with each other to a greater or lesser extent at a given period of time
Biogeome: A set of ecosystems that are similar in their structural and functional organization (e.g. rainforest biogeome, desert biogeome, river ecosystem biogeome or reobiogeome).
Biome: the biotic part of a biogeome, the totality of all its biocoenoses.
Geom: the abiotic part of a biogeome, the totality of conditions, environmental factors, resources. That supports life.
Biospheromeron: the largest element in the structure of the biosphere. There are four of them: surface photic and productive ocean biospheromeron, dysphotic bottom destructive biogeom of the ocean, intermediate inert, surface biospheromeron of continents.
Biocholida: the totality of all living things in the biosphere
Geoholida: the totality of all abiotic conditions, factors, resources that support life in the biosphere
Living matter (in Vernadsky's understanding): the totality of all living organisms of the biosphere in the first approximation similar to Geomerida and Biocholida.
Reviewer: Fourth, the figures used are not easy to follow and the legends are not very explanatory. It is necessary to read everything to understand them. The figures should speak for themselves with the help of the legends.
RESPONSE: Thank you. The legends to the drawings have been made more expanded
Reviewer: Fifth, the history of the biosphere should be explained concerning the geological scale.
RESPONSE: This is a good and useful remark, but the geological and paleontological history of the Earth, as it seems to us, has already been described many times, besides, it will increase the volume of the work, but we will make some remarks regarding the ideas about the change of periods of evolutionary development of the biosphere (although, in the text there is a brief mention and description of both the traditional geochronological division of the Earth's history and less traditional one, according to the concept of G. Zavarzin).
Reviewer: Sixth, explain Vernadsky's ideas since the literature cited is not accessible.
RESPONSE: To the cited references to English-language (available literature) we add the following, our own contemporary articles, which reflect both some little-known biographical data of Vernadsky and conceptual provisions of his theory
Besides, we additionally give a brief characterization (summary) of V. Vernadsky's concept. This fragment is added to the text:
Vladimir Vernadsky's concept is that the complex of living organisms (he calls it "living matter") interacts with the inert elements of the Earth's geological system, actively influences it and forms the Biosphere as a united self-regulating and self-sustaining system. V. Vernadsky's concept of the connections and development of living matter in the changing conditions of the planet as a whole was like a Copernican revolution in understanding the world (Levit Protasov, 2023ab). It allowed for the first time to look at life and the geological basis of life existence as a single whole, a single system, although earlier, such great minds as J.B.Lamarck, A.Humboldt, E Suess tried to move from particular knowledge to planetary generalizations.
Add to references
- Levit, G.S.; Protasov A.A. Vladimir Vernadsky’s “Copernican Turn”. Ann. Hist. Phil. Biol. 2023а, 27, 43-70 https://doi.org/10.17875/gup2023-249-84
- Levit G.S., Protasov A. (2023б) Living Matter: A Key Concept in Vladimir Vernadsky’s Biogeochemistry. Forum Interdisziplinäre Begriffsgeschichte, 1(12): 9-22
Reviewer: Details:
The concept of no change should be understood as stasis (see Eldredge & Gould 1972).
RESPONSE: The absence of changes (and we point it out on the example of Mesozoic vegetation, according to V.Krasilov's works) is conditional, it can be considered as an "external characteristic" of ecosystems and the whole appearance of the biosphere, because STASIS cannot generate MOTION, periods of coherent evolution cannot be considered as complete, absolute stasis. Here there is a peculiarity of the "Russian paradigm" (Lekiavicius, 2003): in Cyrillic literature stasis has never been considered as a stop of motion, but rather some stabilization of form with an internal complex movement of content.
Reviewer: The principle of the pressure of life must consider extinctions. The diversity of life is not static nor does it have growth with some trends due to large extinctions (see Futuyma & Kirkpatrick 2017). Diversification = speciation-extinction.
RESPONSE: Thanks for the comment, we think that extinction is one of the consequences of the pressure of life. The pressure of life both "cleans" the ecological space and fills the free space with new biotic content.
Reviewer: Improving the selection principle, the struggle of the fittest is not all that Darwin proposed (see Futuyma & Kirkpatrick 2017).
RESPONSE: Thanks for the comment, we have made the following addition to the text:
We draw attention to the fact that the concept of natural selection and the struggle for existence are far from being the only provisions of the theory of origin of species as interpreted by C. Darwin (Futuyma & Kirkpatrick 2017, Krasilov 1986)
Add to References
Futuyma, J. D., & Kirkpatrick, M. (2017). Evolution (4th Ed.). sinauer, Sunderland, MA: Sinauer Associates.
Reviewer: The continuity principle must consider mass extinctions. Life on Earth is not continuous because of mass extinctions.
RESPONSE: Respected Reviewer raised a very important question about the controversy of ambiguity of the continuity concept. In our work, which we mention in the paper (Protasov, A. A.; Uzunov, Y. I.; Sylaieva, A. A.; Gromova, Yu.; et al.. Ecological continuum: fundamental concepts and use in applied hydrobiology. Hydrobiological Journal 2022, 58(5), 3-15 https://doi.org:10.1615/HydrobJ.v58.i5.10) it is precisely a certain system of concepts/sub- concepts of continuum in ecology that is considered. Indeed, "from the point of view" of trilobites or dinosaurs, life is not continuous, finite, but the phenomenon of biopoesis should be considered in the geological time scale as a single phenomenon, due to which the existence of life as such, the existence of living things up to the present time could only be on an absolutely continuous, uninterrupted basis. The life of the biosphere is also continuous, but the life/existence of individual species or even biogeomes has a beginning and an end, this is the morphoprocess, i.e. the continuum of the whole with a constant change of its parts.
RESPONSE: We've made an addition to the text of the article:
If we consider Punctuated evolution (Eldredge Gould. 1972) as an alternative to a purely gradualist approach, then we cannot consider it as an evolutionary discontinuity of the biosphere as a whole, including the biocholida .
Extremely grateful, the literature cited by the reviewer has been added to the list in the article
Eldredge, N., & Gould, S. J. (1972). Punctuated equilibria: an alternative to phyletic gradualism. Models in paleobiology, 82, 115.
Reviewer: Change taxonomic to phylogenetic. Taxonomy is a human activity of classification that is not an evolutionary property, but phylogeny of lineages is.
RESPONSE: We agree with the respected Reviewer that Taxonomy, as an ordering of the richness of life forms, is to a large extent a product of human creative activity. However, it is impossible to deny that taxonomy is connected with the real diversity of forms linked by the history of origin and development. In this sense, it fulfills a sign-information function that helps to reflect the real richness of life in a coherent for all/unified sign system. There is a certain tradition to understand phylogeny as the evolution of "large taxa" (e.g., animal types), which, however, are in turn part of the general taxonomy. Therefore, a strict separation of "taxonomic" and "phylogenetic" approaches in thinking about evolution is not always possible. In this case, in our work we deal with real biotic diversity using the taxonomic approach.
Reviewer: The ontogenetic stages of systems I,II,III,IV must be explained in terms of the geological scale. When they occur, when they begin, and when they end (Fig. 1, 2).
RESPONSE: We must make a clarification for the honorable Reviewer (also clarification made in the text): Figures 1 and 2 are related by a common concept, but different in the essence of the examples.
Their similarity is that on the example of ontogenesis of some organism (Fig. 1) we show the existence of different slices of states, in a given period of time and their combinations. The picture of these combinations creates a picture characteristic of this state. For example, the state "childhood" cannot be characterized by a large development of the generative trend. Neither can the state of "old age" be characterized by the decline of this trend. The development of some technical objects, architecture, etc., any systems that develop in one way or another, can be illustrated in a similar way. For example, for the development of technical means of transportation, the following trends will be important: materials, energy source, energy converter, functional needs and, strangely enough, a purely human factor - fashion.
For the evolutionary system of the biosphere, we present several states that are close to the geochronological and paleontological schemes. Fig. 2 reflects, rather conventionally, that the main trends (we propose 9, their number for different approaches may differ) have different "power" in different states of the entire evolutionary system, which creates a unique picture of the appearance of each state.
In this work, we did not aim to directly link the general, fundamental scheme of the evolutionary system of the biosphere with the known geochronology of the development of the planet and terrestrial life. This is the task of future research.
Reviewer: Finally, I think it is a very good review of ideas unknown to student readers and researchers in the rest of the world and can create a bridge between world knowledge.
RESPONSE: We are extremely grateful to the distinguished Reviewer for supporting our efforts in the field not only to express our thoughts and ideas, but also to draw the attention of our colleagues to the issue of the need for an active and productive exchange of scientific results
Reviewer: Recommended Literature
Futuyma, J. D., & Kirkpatrick, M. (2017). Evolution (4th Ed.). sinauer, Sunderland, MA: Sinauer Associates.
Eldredge, N., & Gould, S. J. (1972). Punctuated equilibria: an alternative to phyletic gradualism. Models in paleobiology, 82, 115.
RESPONSE: Extremely grateful, the literature cited by the reviewer has been added to the list in the article

Round 2
Reviewer 1 Report
Comments and Suggestions for Authors
The authors took my comments into account and made the necessary corrections. Additionally, inclusions were also made, with which I agree. I accept the arguments proposed by the authors regarding the structure and functioning of the term biogeome. I consider that the new revised version of the article can be recommended for publication in the journal in the form presented by the authors.
Author Response
Dear Editor and the Reviewer 1,
Thank you for the paper review and comments.
With best regards,
Prof Sophia Barinova,
corresponding author
Reviewer 2 Report
Comments and Suggestions for Authors
Dear authors
I am very pleased with the quick response and the great improvement of the manuscript and both figures. I think it is essential that when this manuscript is published it is widely disseminated in social networks. I will include it in my Evolution classes at my university.
Congratulations for spreading these ideas.
Best regards
Author Response
Dear Editor and the Reviewer 2,
Thank you very much for the paper review and comments.
With best regards,
Prof Sophia Barinova,
corresponding author